# Synthetic Vasculature and Pathology Enhance Vision-Language Model Reasoning

**Chenjun Li**[1,2]                                           CL2733@CORNELL.EDU
**Cheng Wan**[1,2]                                            CW2222@CORNELL.EDU
**Laurin Lux**[2,3]                                           LAURIN.LUX@TUM.DE
**Alexander H. Berger**[2,3]                                  A.BERGER@TUM.DE
**Richard B. Rosen**[4]                                       RROSEN@NYEE.EDU
**Martin J. Menten**[3]                                       MARTIN.MENTEN@TUM.DE
**Johannes C. Paetzold**[2,5]                                 JPAETZOLD@MED.CORNELL.EDU

[1] *School of Electrical and Computer Engineering, Cornell University, Ithaca, NY 14853, USA*

[2] *Weill Cornell Medicine, New York, NY 10021, USA*

[3] *School of Computation, Information and Technology, Technical University of Munich, 80333 Munich, Germany*

[4] *New York Eye and Ear Infirmary of Mount Sinai, New York, NY*

[5] *Cornell Tech, New York, NY 10044, USA*

**Editors:** Accepted for publication at MIDL 2026

## Abstract

Vision-language models (VLMs) offer a promising path toward interpretable medical diagnosis by allowing users to ask about clinical explanations alongside predictions and across different modalities. However, training VLMs for detailed reasoning requires large-scale image-text datasets. In many specialized domains, for example in reading optical coherence tomography angiography (OCTA) images, such precise text with grounded description of pathologies is scarce or even non-existent. To overcome this bottleneck, we introduce synthetic vasculature reasoning (SVR), a framework that controllably synthesizes images and corresponding text, specifically: realistic retinal vasculature with diabetic retinopathy (DR) features: capillary dropout, microaneurysms, intraretinal microvascular abnormalities, and tortuosity, while automatically generating granular reasoning texts. Based on this we curate OCTA-100K-SVR, an OCTA image-reasoning dataset with 100,000 pairs. Our experiments show that a general-purpose VLM (Qwen3-VL-8b) trained on the dataset achieves a zero-shot balanced classification accuracy of 86.69% on real OCTA images, demonstrating performance comparable to supervised baselines. Through human expert evaluation we also demonstrate that it significantly enhances explanation quality and pathology localization on clinical data.

**Keywords:** VLM, CoT, OCTA, DR, Synthetic Pathology

## 1. Introduction

Vision-language models (VLMs) integrate visual processing with natural language reasoning. Many studies discuss how such models can be particularly valuable in medical image analysis, as they can move beyond simple classification to support interpretable diagnosis (Li et al., 2023; Zhang et al., 2024; Yang et al., 2025; Sellergren et al., 2025). Compared to traditional black-box classifiers, VLMs could explicitly describe *what* features are present,

*where* pathologies are located, and *why* a specific diagnosis is suggested. This explainability is critical for clinical validation.

Chain-of-thought (CoT) reasoning enables models to generate step-by-step explanations that simulate human workflows (Wei et al., 2022). In medical imaging, this approach breaks down diagnosis into interpretable stages: identifying anatomical structures, detecting and localizing abnormalities, and synthesizing these findings into a conclusion. However, training VLMs to produce clinically accurate CoT requires extensive datasets with annotations that exceed standard image-level labels (Pan et al., 2025; Lai et al., 2025). Privacy regulations, significant acquisition costs, and the requirement for specialized expertise restrict data availability, particularly for modalities like optical coherence tomography angiography (OCTA), where even the largest public datasets that include OCTA typically involve only hundreds to a few thousand subjects (Li et al., 2024; AI-READI Consortium, 2024). While a recent approach has attempted to improve VLM interpretability on OCTA images via graph-based knowledge extraction (Li et al., 2025), it is limited to small sample sizes and lacks detailed localization of different pathological features. Without ground-truth annotations, it is difficult to train VLMs that provide high-quality, location-specific explanations.

Synthetic data generation offers a practical solution for data augmentation in VLM training (Ma et al., 2025; Wu et al., 2025). Synthetic generation provides control over visual features and automatically produces ground-truth annotations. Despite these benefits, synthetic data is underutilized in medical VLM training. Previous work by Kreitner et al. (2024) validated synthetic OCTA for vessel segmentation but was limited to healthy vasculature, without pathological features and text-based reasoning chains. A framework capable of generating both realistic pathology and corresponding grounded explanations is needed to enable VLMs to diagnose, understand and reason about OCTA images.

We investigate this approach for diabetic retinopathy (DR) staging using OCTA images. DR is a primary cause of vision loss (Lee et al., 2015), and accurate diagnosis of DR relies on the detection and localization of microvascular abnormalities (Alam et al., 2020; Sun et al., 2021). Current public datasets (Dai et al., 2021; Li et al., 2024; AI-READI Consortium, 2024) generally provide only image-level labels.

In this work, we introduce **Synthetic Vasculature Reasoning (SVR)**, a pathology-aware OCTA synthesis framework designed to improve VLM reasoning capabilities (Code and dataset available at: https://github.com/d0ng231/OCTA-SVR). Our work makes the following contributions:

- We developed the first module capable of simulating four distinct DR features on synthetic OCTA images: capillary dropout, microaneurysms, IRMA, and vessel tortuosity.

- Based on the controllable features, we present **OCTA-100K-SVR**, a synthetic dataset with 100,000 pairs of high-quality OCTA images and texts for VLM training, and demonstrate that scaling synthetic training data improves VLM performance in both classification and reasoning.

- Through evaluations on proprietary and public datasets, we show that VLMs trained via SVR generate clinically accurate explanations and correct pathology localization, reaching the performance of other models **without fine-tuning on real data**.

## 2. Method

### 2.1. Overview

In Fig. 1 we illustrate the core of our method, which is based on a simulation module that generates topologically accurate vascular graphs with controlled diabetic retinopathy features. A generative adversarial network (GAN) then converts vessel maps into realistic OCTA images, while a teacher VLM converts the corresponding pathology metadata into grounded Chain-of-Thought reasoning texts. By aligning synthetic OCTA images with granular reasoning across 100,000 samples, the framework enables the VLM to learn robust diagnostic representations prior to fine-tuning on limited real-world clinical data.

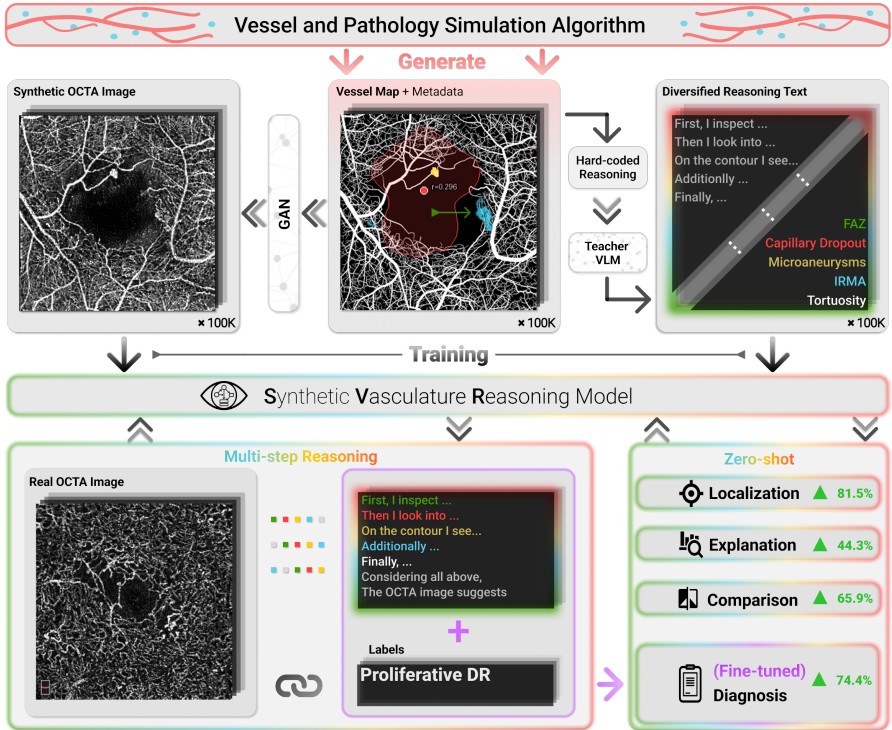

Figure 1: **Overview of the Synthetic Vasculature Reasoning Framework.** The pipeline begins with the simulation of vessels and pathology. In this way we generate ground-truth vessel maps with and without various DR hallmarks (detailed in Sec. 2.2 and Fig. 2). These maps are then used for: (1) visual synthesis, where a pretrained GAN from Kreitner et al. (2024) converts maps into realistic synthetic OCTA images, and (2) text generation, where structured pathology metadata is converted into diversified Chain-of-Thought reasoning text using GPT-5 (OpenAI, 2025) (see Pseudo code 1 for more details). On these image-text pairs we can fine-tune VLMs which are capable of multiple tasks including disease staging, locating pathological features, and explaining their reasoning. The marked improvements in localization, explanation, comparison and diagnosis are based on expert evaluation (see Sec. 4).

## 2.2. Synthetic Vessel Growth and Pathology

**Synthetic OCTA generation pipeline.** We build on the statistical angiogenesis simulator of Kreitner et al. (2024), which itself adapts the space–colonization model of Rauch and Harders (2021). In brief, the retinal vasculature is represented as a forest of rooted 3D binary trees growing in a normalized box $\Omega = [0,1] \times [0,1] \times [0, h_z]$, where the lateral coordinates $(x, y) \in [0,1]^2$ correspond, after global scaling, to a $3 \times 3\,\mathrm{mm}^2$ macular crop and $h_z$ denotes the normalized slab thickness in the axial ($z$) direction. Arterial and venous trees are grown in two successive layers that together form the superficial and deep vascular complexes, driven by randomly sampled oxygen sinks and $CO_2$ sources. Growth follows local attraction cones and Murray's law–based bifurcation rules, producing capillary beds and major vessels. After growth, the vessel graph is voxelized into a 3D volume, projected to an en-face map, and passed through a GAN-based contrast adaptation module to produce realistic synthetic OCTA images.

To increase variability in viewpoint and morphology, we add a random shift and jitter of the foveal avascular zone (FAZ) within the normalized image plane before projection. All variables and parameters of the growth model are described in Appendix A.

**Pathology-aware graph augmentation.** Going beyond previous works which have been focused on healthy vessel graph generation (Menten et al., 2022; Kreitner et al., 2024; Wittmann et al., 2024; Prabhakar et al., 2024), we introduce a pathology module that operates directly on the 3D tree representation. For each synthetic sample we first use the base simulator to complete the growth of arterial and venous forests, and then apply a sequence of pathology-specific graph remodeling operations that are clinically motivated by the four defining pathologies of DR that show in OCTA images of the retinal vasculature (Kaizu et al., 2017; Alam et al., 2020; Sun et al., 2021; Yavari et al., 2026) (See Fig. 2). Although the vasculature is simulated in 3D with nodes $\mathbf{x} = (x, y, z)$, all pathology fields are parameterized in the normalized en-face plane using the lateral coordinates $\tilde{\mathbf{x}} = (x, y) \in [0,1]^2$.

**Capillary dropout.** We first sample $n$ non-perfusion regions around the FAZ. For a region centered at $\mu_k$, its boundary $\tilde{R}_k$ is modeled as an irregular ellipse modulated with a series of angular harmonics and a smooth spatial noise field. We then define a continuous dropout field $c(\mathbf{x}) \in [0,1]$ that encodes how deep the regression should be at each location. Terminal vessel segments (capillaries) inside high-$c(\mathbf{x})$ areas (with threshold $\tau_{\mathrm{reg}}$) are stochastically pruned, with sampling weights that favor small-radius vessels and locations close to the lesion core. In addition, surviving vessels inside dropout regions undergo mild elongation and dilation to mimic remodeling at the borders of ischemic zones. The full definition of $c(\mathbf{x})$ and the pruning/remodeling rules is provided in Appendix A.2.

**Microaneurysms (MA).** Microaneurysms are modeled as short bulbous side branches that bud from existing vessels near dropout borders. For each arterial segment whose endpoint lies in a specified band $b_{MA}$ of the dropout field, we sample a Bernoulli trial whose success probability is modulated by both the global dropout severity and the local value of $c(\mathbf{x})$, making microaneurysms more likely (but not guaranteed) near more severe dropout. Successful trials spawn a small perpendicular branch of length proportional to the simulator step size and with a radius $r_{\mathrm{MA}}$ drawn from a clinically plausible range (typically $20$–$80\,\mu m$). Additional child nodes around the MA center $\mathbf{x}_{\mathrm{MA}}$ create irregular cluster-like

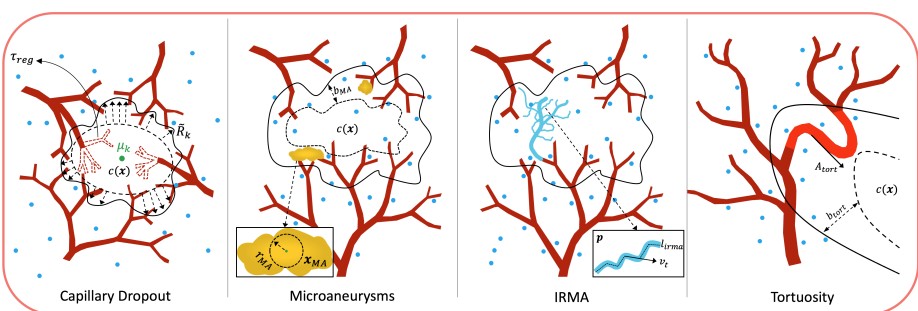

Figure 2: **Illustration of the synthesized DR pathologies.** DR cases show four main pathologies in OCTA images, which we implement in our vessel synthesis model: (1) capillary dropout, (2) microaneurysms (MA), (3) intraretinal microvascular abnormalities (IRMA), and (4) increased tortuosity along dropout borders. Comprehensive mathematical description of the algorithms is provided in Appendix A. Afterwards, the generated (pathology-including) vessel maps are fed to the GAN.

shapes. All MA segments are stored explicitly in the graph and exported as part of the pathology metadata (see Appendix A.3).

**Intraretinal microvascular abnormalities (IRMA).** IRMA refers to abnormal intraretinal remodeling of vessel segments, typically appearing within or near non-perfusion areas. Clinically, IRMA could suggest severe non-proliferative diabetic retinopathy (NPDR) and increased risk of progression, whereas neovascularization (NV) corresponds to new-vessel proliferation characteristic of proliferative diabetic retinopathy (PDR). To model this in a controllable graph simulator, we represent IRMA as short abnormal vessel segments that extend into dropout regions to partially refill regressed capillaries. This can be interpreted as previously existing vessels being remodeled and becoming visible again after capillaries were pruned during dropout synthesis. We identify tips adjacent to dropout borders in the 2D projection and, for a subset selected by a global IRMA severity parameter, simulate short polylines $\mathbf{p}$ of effective length $l_{\mathrm{IRMA}}$ that extend toward the dropout interior. The step direction $\mathbf{v}_t$ combines the previous direction, a weak attraction toward the nearest dropout center, and a small swirling field with random jitter, producing irregular shunt-like paths. We also spawn side branches and apply localized dilation and increased tortuosity along the IRMA segments to match typical OCTA appearance. A detailed description of the polyline construction and radius profiles is given in Appendix A.4.

**Tortuosity along dropout borders.** Finally, we increase vessel tortuosity in a narrow band $b_{tort}$ around the dropout border. For arterial segments whose endpoints lie in this band, we jitter node positions in the direction perpendicular to the local vessel tangent by a small, zero-mean random offset whose amplitude $A_{tort}$ is proportional to a tortuosity gain parameter. This preserves global connectivity and the FAZ geometry while increasing local curvature, producing the characteristic curling of vessels along non-perfusion borders. The exact definition of the tortuosity band and jitter distribution is provided in Appendix A.5.

**Parameterization.** All four pathology types share a small set of interpretable hyper-parameters controlling count, size, strength, and probability. These parameters are sampled

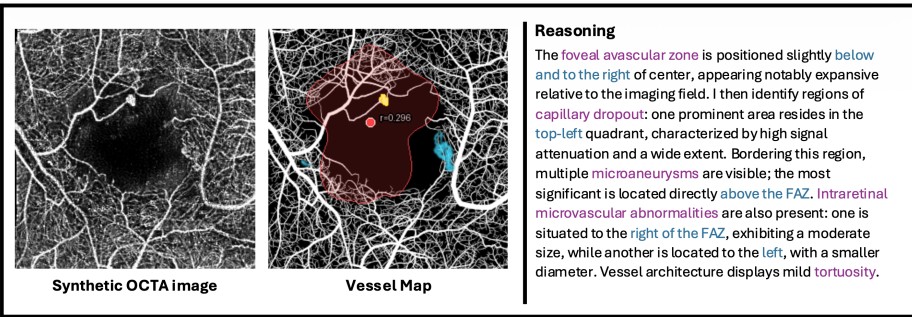

Figure 3: **Synthetic OCTA image and generated reasoning text.** Purple phrases mark structures/pathologies that are directly controlled by the simulator, while the blue texts mark localizations.

from ranges chosen to match clinical OCTA statistics. The full list of parameters and typical values is summarized in Tab. 5 in Appendix A.

## 2.3. VLM Pretraining and Fine-tuning

### 2.3.1. OCTA-100K-SVR Dataset

We refer to the full collection of our 100,000 synthetic image–reasoning pairs as the **OCTA-100K-SVR** dataset. For every simulated sample, the pathology module exports structured metadata describing FAZ geometry and the presence and location of capillary dropout, microaneurysms, IRMA, and tortuosity. From this metadata we first construct a deterministic "template" reasoning paragraph that follows a fixed order (FAZ → dropout → microaneurysms → IRMA → tortuosity) and states explicitly which patterns are present and where they lie relative to the FAZ and macular quadrants.

This template is then passed to a teacher model (GPT-5 (OpenAI, 2025)), which is prompted to rewrite the paragraph while strictly preserving all clinical facts and spatial relations. Here GPT-5 is used only for language-style rewriting, all pathological structures, locations, and disease states are determined by the simulator and its metadata. GPT-5 is chosen primarily due to its stronger instruction-following capability, more stable long-form generation, and more reliable adherence to structured prompts compared to other alternative models we tested in our experiments. It diversifies the Chain-of-Thought descriptions that remain aligned with the metadata (see Fig. 3 and Appendix A.7). For each image we keep one such paragraph as the assistant answer $a$. The user question $q$ is a short prompt asking the model to describe the OCTA image. Training with these image–reasoning pairs minimizes next-token cross-entropy. We use the same conversation format for the in-house OCTA data and add the explicit diagnosis sentence at the end of the reasoning paragraph for the fine-tuning stage.

### 2.3.2. SVR pretraining and fine-tuning

Our main reasoning model, denoted **SVR**, is obtained by pretraining the general-purpose Qwen3-VL-8b-Instruct (Yang et al., 2025) on the synthetic OCTA-100K-SVR dataset. We

treat this as instruction-style supervised fine-tuning on the synthetic image–reasoning pairs while freezing the language module, so that only the vision encoder and multi-modal projection layers are updated. In this stage the model learns OCTA-specific visual features and their alignment to the existing language space without disturbing its general language capabilities. Models trained on smaller synthetic subsets (1k–75k samples) are used in the scaling experiments of Sec. 3.2.

To close the remaining domain gap, we then perform a second stage of supervised fine-tuning on clinical OCTA data, starting from an SVR-pretrained checkpoint. The resulting model is denoted **SVR-FT** and is trained on OCTA images from OCTA-500 and our in-house dataset together with reasoning and final DR labels. In this stage we train the model end-to-end, allowing it to adapt both its decision boundaries and linguistic style to real-world data while retaining the pathology-aware visual grounding acquired from synthetic vasculature. Training details and hyperparameters are summarized in Sec. 3.

## 3. Experiments

### 3.1. Datasets

We evaluate models on one synthetic dataset and two clinical datasets. Apart from OCTA-100K-SVR, the in-house dataset contains 1,286 OCTA deep capillary plexus scans labeled as Healthy, NPDR, or PDR following the 5-fold setup of Lux et al. (2025). For OCTA-500 (Li et al., 2024), we follow the same test split of 189 scans as in Lux et al. (2025) and Li et al. (2025).

### 3.2. Tasks and metrics

We study two DR staging settings. On OCTA-500, models perform binary DR-vs.-Healthy classification. On the in-house dataset, models perform three-label staging (Healthy, NPDR, PDR). In both cases the VLM receives a single OCTA image and a prompt asking about DR staging. We report class-wise precision and recall and use *balanced accuracy* (mean per-class recall) as the primary metric.

We compare SVR-based models with supervised baselines, including ResNet18, a vessel-graph GNN (Lux et al., 2025), and an OCTA-pretrained vision encoder derived from SAM-OCTA (Chen et al., 2025). For the SAM-OCTA baseline, we reuse the OCTA-adapted ViT image encoder and finetune it for DR classification by attaching a lightweight multi-layer perceptron head, without using prompt-based mask decoding. We also compare against several VLM baselines: vanilla Qwen3-VL-8b/30b (Yang et al., 2025), LLaMA-3.2-11B-VL (Touvron et al., 2023), LLaVA-NEXT-8B (Liu et al., 2023), the graph-knowledge-based Qwen3-VL-8b-GFT (Li et al., 2025), and a baseline Qwen3-VL-8b-FT finetuned using only classification labels.

To quantify explanation quality we use GPT-5 as an automatic judge, similar to the ways in Liu et al. (2023) and Li et al. (2025). For each model, GPT-5 scores the candidate response on helpfulness, clinical accuracy, localization accuracy, and relevance, and we report the average score over these dimensions on both the in-house set and a held-out synthetic test set. Two experts in OCTA imaging then each evaluated 30 in-house responses. For each case, experts were presented with six anonymized model outputs in a randomized order

and independently ranked the explanations under the same criteria. In addition to overall ranking, experts marked each sentence in the explanation to indicate whether it correctly referred to a clinically valid retinal region. Based on these sentence-level annotations, we introduce an objective localization metric defined as the number of correct localizations per case, computed by counting localizations with expert-confirmed spatial grounding. To examine whether GPT-5 scores provide a meaningful proxy rather than a stand-alone clinical metric, we perform a scaling effects experiment in which Qwen3-VL-8b-SVR is trained on synthetic subsets from 1k to 100k samples and evaluated on OCTA-500. We track both balanced accuracy and GPT-5 score as functions of synthetic dataset size, and further analyze the correlation between GPT-5 scores and expert ratings to assess their alignment.

Ablation experiments isolate key components of SVR: (i) prompt diversification vs. static template reasoning, and (ii) the contribution of individual pathology types by removing dropout or microaneurysms from the reasoning. We additionally compare against a baseline (Qwen3-VL-8b CoT-FT) that fine-tunes on GPT-5-generated reasoning instead of SVR.

### 3.3. Training setup

All experiments are implemented with LLaMA-Factory (Zheng et al., 2024) on four NVIDIA H100 GPUs. OCTA images are resized to $512 \times 512$. In the SVR stage, we fine-tune Qwen3-VL-8b-Instruct on OCTA-100K-SVR using a cosine learning-rate schedule. To prevent the model from merely memorizing synthetic text patterns, we freeze the language backbone and only update the vision encoder and multimodal projector. In the subsequent SVR-FT stage, we maintain this configuration: the language model remains frozen while the vision components are further adapted to clinical OCTA images with their reasoning and labels. Additional implementation details are provided in Appendix C.

## 4. Results and Discussion

### 4.1. DR Staging

#### 4.1.1. Zero-shot and supervised fine-tuning

Tabs. 1 and 6 summarize DR staging performance on OCTA-500 and the in-house dataset. Directly fine-tuning a general-purpose VLM on limited classification labels (*Qwen3-VL-8b-FT*) is unstable and causes mode collapse, where the DR cases are neglected. In contrast, SVR pretraining (*Qwen3-VL-8b-SVR*) already produces comparable balanced accuracy in a zero-shot setting, despite never seeing real scans.

When subsequently fine-tuned on clinical data (*Qwen3-VL-8b-SVR-FT*), the model achieves consistently higher and more balanced recall across all stages, especially for advanced DR on both datasets. Compared with purely supervised CNN and GNN baselines, SVR-FT reaches similar or better balanced accuracy while additionally producing explanations. This suggests that SVR pretraining provides a strong reasoning capability regularizes downstream training, improves sensitivity to disease, and mitigates overfitting.

#### 4.1.2. Scaling effects of synthetic data

The scaling experiment in Tab. 2 and Fig. 5 shows that increasing the size of OCTA-100K-SVR improves both zero-shot DR classification and GPT-5 scores. With very small

Table 1:  DR staging on OCTA-500. H = Healthy, DR = Diabetic Retinopathy.

| Model | Prec(H) | Prec(DR) | Rec(H) | Rec(DR) | Bal.Acc |
|---|---|---|---|---|---|
| ResNet18 (Lux et al., 2025) | 0.8734 | 0.2903 | 0.8625 | 0.3103 | 0.5864 |
| SAM-OCTA (Chen et al., 2025) | 0.9296 | 0.4043 | 0.8250 | 0.6552 | 0.7401 |
| GNN (Lux et al., 2025) | 0.9636 | 0.9583 | 0.9938 | 0.7931 | 0.8934 |
| Qwen3-VL-8b (Yang et al., 2025) | 0.8503 | 0.5000 | 0.9938 | 0.0345 | 0.5142 |
| Qwen3-VL-30b (Yang et al., 2025) | 0.8511 | **1.0000** | **1.0000** | 0.0345 | 0.5173 |
| Qwen3-VL-8b-GFT (Li et al., 2025) | 0.9718 | 0.5846 | 0.9347 | 0.7703 | 0.8525 |
|  | ± 0.0073 | ± 0.0408 | ± 0.0091 | ± 0.0594 | ± 0.0275 |
| Qwen3-VL-8b-FT | 0.8551 | 0.2000 | **1.0000** | 0.0621 | 0.5310 |
|  | ± 0.0189 | ± 0.4000 | ± 0.0000 | ± 0.1241 | ± 0.0620 |
| Qwen3-VL-8b-SVR (ours) | 0.9667 | 0.6154 | 0.9063 | 0.8276 | 0.8669 |
| Qwen3-VL-8b-SVR-FT (ours) | **0.9745** | 0.7693 | 0.9488 | **0.8621** | **0.9054** |
|  | ± 0.0096 | ± 0.1180 | ± 0.0323 | ± 0.0546 | ± 0.0256 |

synthetic subsets, performance is highly variable and remains close to chance level, indicating that limited coverage of vascular topologies and lesion patterns is insufficient for robust reasoning. Once the synthetic dataset reaches tens of thousands of samples, we observe a sharp improvement.

Table 2:  Scaling Effects of Synthetic Data (zero-shot test on OCTA-500).

| Size | Prec(H) | Prec(DR) | Rec(H) | Rec(DR) | Bal.Acc | GPT-5 Score |
|---|---|---|---|---|---|---|
| 0 | 0.8503 | 0.5000 | 0.9938 | 0.0345 | 0.5142 | 50.3 |
| 1k | 0.8421 | 0.1467 | 0.6000 | 0.3793 | 0.4897 | 58.6 |
| 5k | 0.8595 | 0.1765 | 0.6500 | 0.4138 | 0.5319 | 54.2 |
| 10k | 0.8672 | 0.1967 | 0.6938 | 0.4138 | 0.5538 | 55.7 |
| 25k | 0.8951 | 0.3043 | 0.8000 | 0.4828 | 0.6414 | 61.8 |
| 50k | 0.9241 | 0.5484 | 0.9125 | 0.5862 | 0.7494 | 67.3 |
| 75k | 0.9608 | **0.6389** | **0.9188** | 0.7931 | 0.8559 | **73.2** |
| 100k | **0.9667** | 0.6154 | 0.9063 | **0.8276** | **0.8669** | 72.6 |

### 4.1.3. Ablation Studies

The ablation results in Tab. 3 highlight two central components of SVR. First, removing reasoning diversification (w/o diversifying) degrades downstream performance. This indicates that varied but fact-preserving CoT texts are critical for preventing the VLM from memorizing a fixed template and instead encouraging it to condition genuinely on the image content. Second, ablating individual pathology types (w/o dropout or microaneurysms) also decreases the performance, showing the importance of explicitly modeling all key DR hallmarks in the pipeline.

Table 3:  Ablation Study.

| Model | Prec(H) | Prec(DR) | Rec(H) | Rec(DR) | Bal.Acc |
|---|---|---|---|---|---|
| Qwen3-VL-8b (Yang et al., 2025) | 0.8503 | 0.5000 | 0.9938 | 0.0345 | 0.5142 |
| Qwen3-VL-8b CoT-FT | 0.8951 | 0.4444 | 0.9063 | 0.4138 | 0.6601 |
| Qwen3-VL-8b SVR (w/o diversifying) | 0.8912 | 0.3095 | 0.8188 | 0.4483 | 0.6335 |
| Qwen3-VL-8b SVR (w/o Dropout) | 0.9281 | 0.5000 | 0.8875 | 0.6207 | 0.7541 |
| Qwen3-VL-8b SVR (w/o MA) | 0.9037 | 0.2963 | 0.7625 | 0.5517 | 0.6571 |
| LLaMA-3.2-11B-VL-SVR (Touvron et al., 2023) | 0.8939 | **1.0000** | **1.0000** | 0.3448 | 0.6724 |
| LLaVA-NEXT-8b-SVR (Liu et al., 2023) | 0.9355 | 0.5588 | 0.9063 | 0.6552 | 0.7808 |
| Qwen3-VL-8b-SVR (ours) | 0.9667 | 0.6154 | 0.9063 | 0.8276 | 0.8669 |
| Qwen3-VL-8b-SVR-FT (ours) | **0.9745** | 0.7693 | 0.9488 | **0.8621** | **0.9054** |

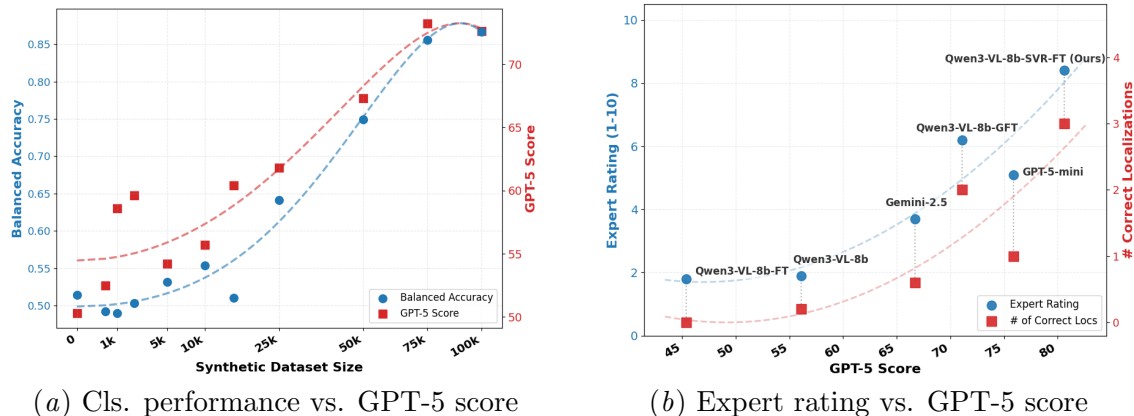

(*a*) Cls. performance vs. GPT-5 score       (*b*) Expert rating vs. GPT-5 score

Figure 4: **Relationship between GPT-5 scores and model performance.** (a) Correlation with DR classification performance. (b) Correlation with expert ratings.

## 4.2. Quality of Explanation

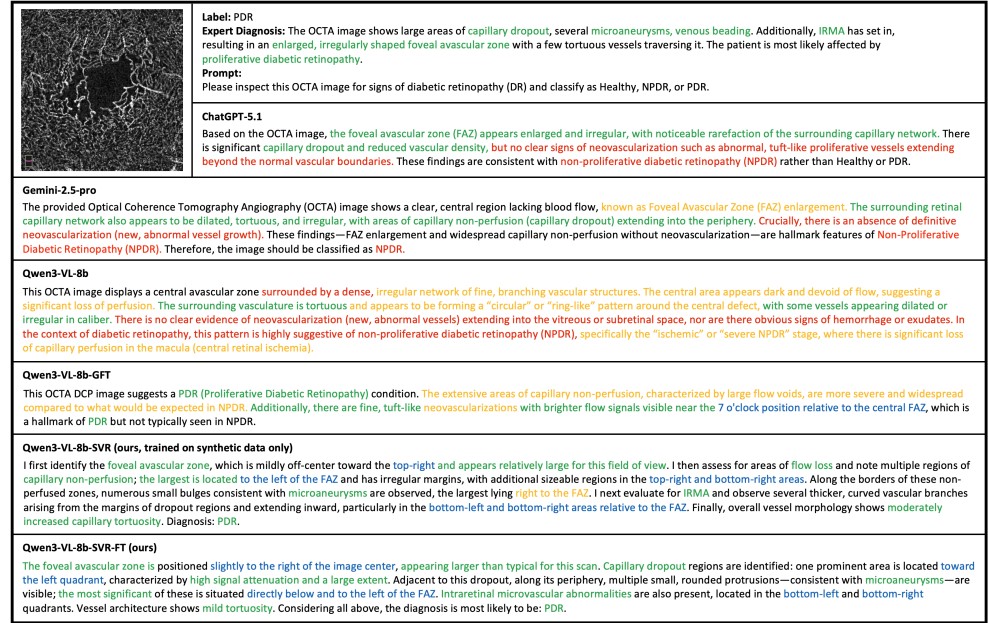

Figure 5: **Comparison of model response.** True, partially true and false sentences are marked in green, yellow and red, respectively.

We next examine explanation quality. Using GPT-5 as an automatic judge, we score each model response on both the in-house and synthetic test sets (Tab. 4), where SVR-FT obtains the highest GPT-5 scores in all settings. Expert ratings are also provided in Tab. 4, where SVR-FT again achieves the best result. Qualitative examples in Fig. 5 show

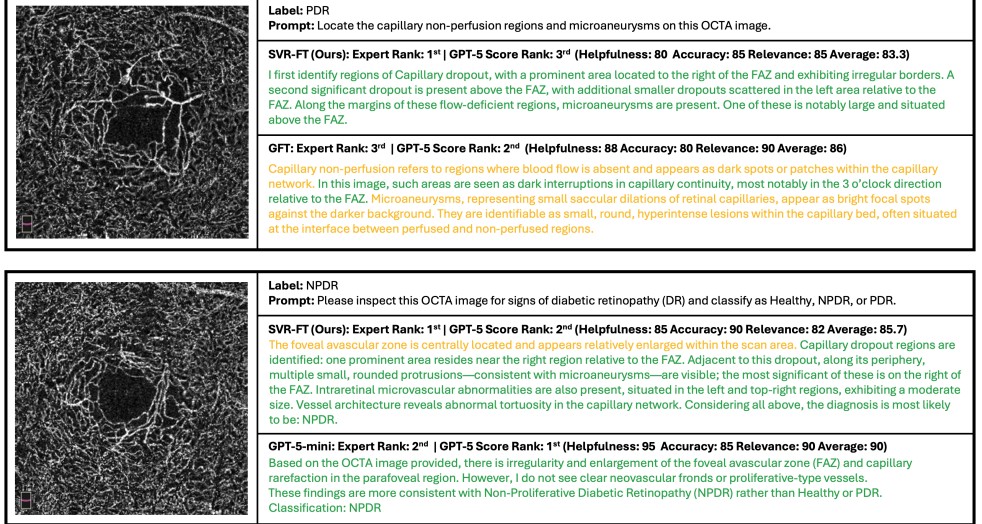

Figure 6: **Failure case analysis of divergence between expert ratings and GPT-5 scores. Case A (Top):** The expert prefers SVR-FT for its specific localization. However, GPT-5 assigns a higher score to the GFT baseline, lower scores in Helpfulness and Relevance to SVR-FT, while rewarding GFT's generic but fluent definitions. **Case B (Bottom):** The expert ranks SVR-FT first for correct identification of microaneurysms and dropout. In contrast, GPT-5 shows a preference for GPT-5-mini, awarding the model from its own family near-perfect Helpfulness scores despite the expert finding it less diagnostically precise.

that SVR-FT produces grounded descriptions that discuss FAZ, dropout, microaneurysms, IRMA, and tortuosity, rather than generic or hallucinated findings.

While GPT-5 scores should not be interpreted as a definitive measure of clinical correctness, they provide a complementary signal for assessing explanation quality when used with appropriate context. In this paper, we treat GPT-5 scores as an auxiliary evaluation metric rather than a ground-truth clinical benchmark. In addition to the correlation analysis between GPT-5 scores and DR classification performance (Fig. 4(a)), we further analyze the relationship between GPT-5 scores and expert ratings (Fig. 4(b)). We observe a strong positive alignment in both analyses, suggesting that although imperfect, GPT-5 scores capture meaningful aspects of explanation quality when interpreted alongside expert judgment rather than in isolation.

We also analyzed failure cases where GPT-5 scores disagreed with expert rankings (Fig. 6). The discrepancies show that GPT-5 scores can be biased by linguistic styles and fluency, and thus must be interpreted alongside expert review to ensure clinical utility.

### 4.3. Extension of the Simulation Mechanism to Future Work

While this work focuses on DR, the SVR framework is built upon a parametric simulator that is not inherently limited to a single pathology. The core growth mechanism is driven

Table 4: Explanation quality evaluated by GPT-5 and human experts in ophthalmology (higher is better). Metrics: H = helpfulness, A = accuracy (localization & clinical), R = relevance, Avg = average score. The last column indicates the average number of expert-verified correct pathological localizations per case.

| Model | In-house Dataset | | | | Synthetic Test Set | | | | Expert Rating (1–10) | # of correct locs. |
|---|---|---|---|---|---|---|---|---|---|---|
| | H | A | R | Avg | H | A | R | Avg | | |
| GPT-5-mini (OpenAI, 2025) | 76.2 | 68.7 | 82.9 | 75.9 | 73.4 | 65.1 | 79.2 | 72.6 | 5.1 | 1.0 |
| Gemini-2.5-flash (Comanici et al., 2025) | 69.7 | 52.1 | 78.3 | 66.7 | 46.8 | 37.9 | 53.7 | 46.1 | 3.7 | 0.6 |
| Qwen3-VL-8b (Yang et al., 2025) | 56.0 | 36.8 | 75.5 | 56.1 | 58.2 | 45.2 | 75.2 | 59.5 | 1.9 | 0.2 |
| Qwen3-VL-8b-FT | 40.1 | 37.1 | 59.0 | 45.4 | 43.4 | 47.8 | 57.2 | 49.5 | 1.8 | 0.0 |
| Qwen3-VL-8b-GFT (Li et al., 2025) | 71.3 | 67.9 | 74.2 | 71.1 | 72.5 | 69.4 | 75.1 | 72.3 | 6.2 | 2.0 |
| Qwen3-VL-8b-SVR-FT (ours) | **80.6** | **76.2** | **84.9** | **80.6** | **76.8** | **73.5** | **80.4** | **76.9** | **8.4** | **3.0** |

by oxygen sinks, Murray's law bifurcation, and local perfusion constraints, which provides a reusable vascular baseline that mimics general retinal physiology (Kreitner et al., 2024). Because disease-specific features are implemented as modular graph remodeling operations parameterized by explicit variables such as dropout radius ($r_{drop}$), lesion strength ($s_k$), and microaneurysm density ($p_0$), the framework can be extended to other retinal diseases simply by retuning these modules or introducing new geometric primitives. For instance, retinal vein occlusion (RVO) could be simulated by increasing the tortuosity gain ($g_{tort}$) and applying sector-specific dropout masks (Kashani et al., 2015), while glaucoma-like defects could be modeled by adjusting the dropout field to target the peripapillary region and reducing global vessel density ($f_{drop}$) (Spaide et al., 2018). **Importantly, the primary objective of SVR is not merely to maximize the realism of individual synthetic samples, but to generate a database of sufficient scale and parametric diversity to comprehensively cover the spectrum of pathological manifestations**. By mapping these structural parameters to varied disease labels and updating the metadata-to-text templates, SVR transforms empirical clinical knowledge into large-scale, effective instruction-tuning data for generalist medical VLMs.

## 5. Conclusion

In this work, we presented a novel framework for training medical VLMs using synthesized images and pathological features. By simulating realistic retinal vasculature and specific DR pathologies we generated a large-scale synthetic dataset with precise ground-truth annotations and reasoning chains. Our experiments demonstrate that pretraining on this synthetic vasculature (SVR) substantially enhances VLM performance, allowing general-purpose models to outperform specialized supervised methods in diagnostic accuracy even without training on real clinical data, while providing interpretable, clinically meaningful explanations. Importantly, we showed that synthetic scaling could overcome the data scarcity bottleneck in medical imaging, transforming empirical knowledge into highly effective instruction-tuning data. Future work could extend this synthesis approach to 3D volumetric reasoning and multi-modal integration.

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

## Appendix A. Details of Vessel and Pathology Simulation

This appendix provides the mathematical details of the vessel growth and pathology simulation summarized in Sec. 2.2. All coordinates are defined in a normalized 2D en-face domain $[0, 1]^2$ unless otherwise stated, and radii in millimeters are mapped to pixel units via the same physical scale used by the baseline OCTA simulator (Kreitner et al., 2024).

### A.1. Baseline healthy vessel simulation

We adopt the angiogenesis-based statistical growth model of Kreitner et al. (2024), which represents the vasculature as a forest of rooted binary trees growing in a 3D box

$$\Omega = [0, 1] \times [0, 1] \times [0, h_z]. \tag{1}$$

Each vessel segment is an edge

$$e = (\mathbf{x}_i, \mathbf{x}_j, r_{ij}), \tag{2}$$

where $\mathbf{x}_i, \mathbf{x}_j \in \Omega$ are the 3D coordinates of the incident nodes and $r_{ij} > 0$ is the segment radius. Sibling segments satisfy Murray's law with bifurcation exponent $\kappa$,

$$r_{\text{parent}}^{\kappa} = r_{\text{child},1}^{\kappa} + r_{\text{child},2}^{\kappa}. \tag{3}$$

Growth proceeds in two phases (superficial and deep vascular complexes) by repeatedly sampling oxygen sinks (for arteries) and $CO_2$ sources (for veins) and letting leaf or internodes sprout if the attraction points fall into a perception cone of distance $\delta$ and angle $\gamma$ around the current segment. The growth direction is a weighted combination of the mean attraction vector and an optimal branching vector that minimizes deviation from the parent vessel.

**FAZ center shift**   A circular exclusion zone around the FAZ with radius $r_{\text{FAZ}}$ prevents sink placement in the foveal center. To increase variability, we randomly jitter the FAZ center by a vector $\Delta\mathbf{c}$ sampled from a disk of radius $r_{\text{jitter}}$ and clamp the resulting center to a maximum normalized displacement $|\Delta\mathbf{c}| \leq r_{\text{max}}$. This effectively shifts the view while leaving the underlying vasculature unchanged.

After growth, the vessel graph is voxelized at high resolution and projected along the depth axis to obtain a grayscale vessel map $X$ and a binarized segmentation label $\text{bin}(X)$. A pretrained GAN then converts $X$ into a realistic OCTA image.

### A.2. Capillary dropout field and vessel regression

**Dropout regions and inside-score.**   Each dropout lesion $k$ is specified by:

- center $\boldsymbol{\mu}_k = (c_{k,x}, c_{k,y}) \in [0, 1]^2$,

- base radius $r_k > 0$ in normalized units,

- axis ratios $a_k, b_k > 0$ controlling ellipticity,

- harmonic amplitudes $A_{k,m}$ and phases $\phi_{k,m}$ for a small set of modes $\mathcal{H}_k$ (typically $m \in \{2, 3, 5\}$),

- shape exponent $\alpha_k > 0$,

- noise gain $g_k \geq 0$ and lesion strength $s_k \in [0, 1]$.

For a location $\mathbf{x} = (x, y)$ we define its coordinates relative to lesion $k$ as

$$\boldsymbol{\delta}_k(\mathbf{x}) = \mathbf{x} - \boldsymbol{\mu}_k, \quad \rho_k(\mathbf{x}) = \|\boldsymbol{\delta}_k(\mathbf{x})\|_2, \quad \theta_k(\mathbf{x}) = \mathrm{atan2}(\delta_{k,y}, \delta_{k,x}). \tag{4}$$

The radius of the underlying ellipse in direction $\theta$ is

$$R_k(\theta) = \frac{r_k}{\sqrt{\left(\frac{\cos\theta}{a_k}\right)^2 + \left(\frac{\sin\theta}{b_k}\right)^2}}, \tag{5}$$

and we modulate it with angular harmonics to obtain an irregular boundary

$$\tilde{R}_k(\theta) = R_k(\theta) \left[ 1 + \sum_{m \in \mathcal{H}_k} A_{k,m} \cos(m\theta + \phi_{k,m}) \right]. \tag{6}$$

We use the notation $[z]_+ = \max(z, 0)$. The geometric "inside-score" of lesion $k$ is

$$u_k(\mathbf{x}) = \left[ 1 - \frac{\rho_k(\mathbf{x})}{\tilde{R}_k(\theta_k(\mathbf{x}))} \right]_+^{\alpha_k}, \tag{7}$$

which smoothly decays from 1 at the center to 0 at the boundary.

To introduce additional boundary irregularity we construct a smooth noise field $n_k(\mathbf{x}) \in [0, 1]$ by summing a few sinusoidal components in $x$ and $y$. With a noise gain $g_k$ we define

$$c_k(\mathbf{x}) = u_k(\mathbf{x}) \, \mathrm{clip}\big(0.75 + g_k(n_k(\mathbf{x}) - 0.5), 0, 1.2\big), \tag{8}$$

where $\mathrm{clip}(\cdot)$ clamps to the indicated range, and $c_k(\mathbf{x})$ is subsequently capped to $[0, 1]$.

The global dropout field is given by the maximum over lesions,

$$c(\mathbf{x}) = \max_k c_k(\mathbf{x}) \in [0, 1], \tag{9}$$

and the overall dropout severity by

$$s_{\mathrm{max}} = \max_k s_k \in [0, 1]. \tag{10}$$

**Probabilistic pruning of capillaries.** Let $\mathcal{L}$ denote the set of leaf nodes (terminal segments) in one vascular forest. Each leaf $i \in \mathcal{L}$ has position $\mathbf{x}_i$ and radius $r_i$, and we evaluate $c_i = c(\mathbf{x}_i)$. We only consider leaves with

$$c_i \geq \tau_{\mathrm{reg}}, \tag{11}$$

where $\tau_{\mathrm{reg}} \in (0, 1)$ is a regression threshold (we use $\tau_{\mathrm{reg}} \approx 0.35$).

We target a global removal fraction

$$f_{\mathrm{drop}} \approx s_{\mathrm{max}}, \tag{12}$$

with empirical lower/upper bounds to avoid trivial cases. Among eligible leaves we define sampling weights

$$w_i \propto (1 - c_i)^\gamma \, r_i^{-\alpha}, \quad i \in \mathcal{L}, \tag{13}$$

where $\gamma > 0$ biases removal towards the lesion core and $\alpha \in [0.3, 1]$ controls the preference for smaller vessels. We sample $\lfloor f_{\mathrm{drop}}|\mathcal{L}| \rfloor$ leaves without replacement according to $w_i$ and delete the corresponding segments from the graph.

**Remodeling: elongation and dilation.** For non-removed nodes we model subtle remodeling. Let node $i$ with parent $p$ have positions $\mathbf{x}_i$ and $\mathbf{x}_p$, local dropout $c_i = c(\mathbf{x}_i)$, direction

$$\mathbf{d}_{pi} = \mathbf{x}_i - \mathbf{x}_p, \tag{14}$$

and original radius $r_i$. We sample an elongation factor $e_i \in [e_{\min}, e_{\max}]$ and apply

$$\mathbf{x}_i^{(\text{new})} = \mathbf{x}_p + \left[1 + (e_i - 1)c_i\right] \mathbf{d}_{pi}, \tag{15}$$

which increases segment length more strongly near the lesion center.

Similarly, we dilate radii according to

$$r_i^{(\text{new})} = r_i \left[D_{\min} + (D_{\max} - D_{\min})c_i\right], \tag{16}$$

with $D_{\min}, D_{\max} \geq 1$ and optional global radius clamps.

### A.3. Microaneurysm synthesis

Microaneurysms are modeled as short, roughly circular side branches emerging near dropout borders.

**Spawn region.** For each non-root arterial node $i$ with parent $p$ we compute $c_i = c(\mathbf{x}_i)$. If MAs are restricted to dropout borders, we require

$$c_{\min}^{\text{MA}} \leq c_i \leq c_{\max}^{\text{MA}}, \tag{17}$$

for some band $b_{MA} : (c_{\min}^{\text{MA}}, c_{\max}^{\text{MA}}) \subset (0, 1)$.

**Spawn probability.** For each eligible node we perform a Bernoulli trial with probability

$$p_{\text{MA}}(\mathbf{x}_i) = p_0 \left(1 + \lambda_s s_{\max}\right)\left(1 + \lambda_c c_i\right), \tag{18}$$

where $p_0$ is a base MA density, $\lambda_s$ couples MA counts to global dropout severity, and $\lambda_c$ increases density near dropout borders. We also reweight $p_{\text{MA}}$ by the area of the largest dropout region, so larger lesions tend to host more MAs.

**MA geometry.** We form a short side branch. Let

$$\mathbf{u}_{pi} = \frac{\mathbf{x}_i - \mathbf{x}_p}{\|\mathbf{x}_i - \mathbf{x}_p\|_2} \tag{19}$$

be the parent-to-child direction and

$$\mathbf{u}_\perp = (-u_{pi,y}, u_{pi,x}, 0) \tag{20}$$

a perpendicular unit vector in the en-face plane. With simulator step size $d$ and MA length factor $\ell_{\text{MA}}$ we place the MA center at

$$\mathbf{x}_{\text{MA}} = \mathbf{x}_i + \ell_{\text{MA}} d\, \mathbf{u}_\perp. \tag{21}$$

The MA radius is sampled as

$$r_{\text{MA}} \sim \mathcal{U}\left(r_{\min}^{\text{MA}}, r_{\max}^{\text{MA}}\right), \tag{22}$$

where $r_{\min}^{\text{MA}}$ and $r_{\max}^{\text{MA}}$ are specified in millimeters. Additional child nodes can be sampled in a small disk around $\mathbf{x}_{\text{MA}}$ with slightly reduced radii to create irregular clusters.

### A.4. IRMA

IRMA is represented as short, intraretinal shunt-like segments that extend from existing vessel tips into non-perfusion (dropout) regions, reflecting abnormal remodeling and dilation of pre-existing vessels rather than new-vessel proliferation.

**Tip selection.**   We project arterial segments to 2D, identify leaf tips $j$ with position $\mathbf{x}_j$, tangent direction $\mathbf{u}_j$, and local dropout value $c(\mathbf{x}_j)$. Tips too close to the FAZ or clearly outside dropout are down-weighted. A global IRMA severity parameter $s_{\text{IRMA}} \in [0,1]$ determines the number of IRMA groups $G$ and typical sprout length.

**Main segment growth.**   For each selected tip we initialize a polyline $\{\mathbf{p}_t^{(g)}\}_{t=0}^{T_{\text{main}}}$ with

$$\mathbf{p}_0^{(g)} = \mathbf{x}_j. \tag{23}$$

At iteration $t$ we update

$$\mathbf{p}_{t+1}^{(g)} = \mathbf{p}_t^{(g)} + \ell_{\text{IRMA}}\mathbf{v}_t^{(g)}, \tag{24}$$

where $\ell_{\text{IRMA}}$ is a small step size and $\mathbf{v}_t^{(g)}$ is a unit direction obtained as a weighted combination of:

1. previous direction $\mathbf{v}_{t-1}^{(g)}$ (persistence),

2. a weak radial vector pointing toward the closest dropout center,

3. a low-frequency swirling field and isotropic jitter.

We clamp positions to remain within dropout regions and outside the FAZ.

**Radius profile and side branches.**   Along each polyline we use a linearly tapering radius

$$r_{\text{IRMA}}(t) = (1 - \tau_t)r_{\text{start}} + \tau_t r_{\text{end}}, \quad \tau_t = \frac{t}{T_{\text{main}}}, \tag{25}$$

with $r_{\text{start}}$ proportional to the parent vessel radius and $r_{\text{end}} < r_{\text{start}}$. With probability increasing in $s_{\text{IRMA}}$ we spawn side branches starting from intermediate points. These follow the same growth rule but with shorter maximum length $T_{\text{side}}$.

### A.5. Tortuosity along dropout borders

To model increased tortuosity at dropout borders we jitter node positions perpendicular to the local vessel direction within a band of the dropout field.

For a non-root arterial node $i$ with parent $p$, we define the tangent direction

$$\mathbf{u}_{pi} = \frac{\mathbf{x}_i - \mathbf{x}_p}{\|\mathbf{x}_i - \mathbf{x}_p\|_2}, \tag{26}$$

and choose a perpendicular direction $\mathbf{u}_\perp$ as above. Let $c_i = c(\mathbf{x}_i)$ and specify a tortuosity band $b_{tort}$ where

$$c_{\min}^{\text{tort}} \le c_i \le c_{\max}^{\text{tort}}. \tag{27}$$

Nodes outside this band are left unchanged.

Given a gain parameter $g_{\mathrm{tort}} \in [0,1]$ and simulator step size $d$, we set the jitter amplitude

$$A_{\mathrm{tort}} = 0.35\, g_{\mathrm{tort}}\, d, \tag{28}$$

sample

$$\epsilon_i \sim \mathcal{U}(-A_{\mathrm{tort}}, A_{\mathrm{tort}}), \tag{29}$$

and update the node position as

$$\mathbf{x}_i^{(\mathrm{new})} = \mathbf{x}_i + \epsilon_i\, \mathbf{u}_\perp, \tag{30}$$

clipping back to $[0,1]^2$ if necessary. This leaves the topology and global FAZ shape intact but increases local curvature.

## A.6. Parameter ranges and typical settings

Tab. 5 summarizes the key parameters that control the four DR lesion types and the typical ranges used in the OCTA-100K-SVR dataset. Spatial coordinates are normalized to $[0,1]^2$; radii in millimeters refer to the physical scale of the baseline simulator.

Table 5: Key parameters of the simulator and typical settings used in this work.

| Pathology | Parameter | Typical value / range |
|---|---|---|
| Dropout | # regions $n$ | $[0,6]$ per sample |
| | Radius $r_{\mathrm{drop}}$ | $[0.18, 0.32]$ (normalized) |
| | Lesion strength $s_k$ | $[0.90, 0.99]$ |
| | Gradient exponent $\alpha_k$ | $[2.0, 3.0]$ |
| | Noise gain $g_k$ | $[0.20, 0.40]$ |
| MA | Base density $p_0$ | $\approx 0.03$ (per segment) |
| | Radius $r_{\mathrm{MA}}$ | $[0.01, 0.08]$ mm |
| | Length factor $\ell_{\mathrm{MA}}$ | $[0.3, 0.4]$ (in units of $d$) |
| | Strength coupling $\lambda_s$ | $\approx 15$ |
| IRMA | IRMA probability | $\approx 0.4$ |
| | Severity $s_{\mathrm{IRMA}}$ | $[0.2, 0.7]$ |
| | Footprint radius | $[0.015, 0.07]$ (normalized) |
| | Sprout length $l_{\mathrm{IRMA}}$ | $[3,6]$ growth steps |
| Tortuosity | Gain $g_{\mathrm{tort}}$ | $[0.01, 0.5]$ |
| | Band $[c_{\min}^{\mathrm{tort}}, c_{\max}^{\mathrm{tort}}]$ | $[0.30, 0.75]$ |

## A.7. Extra Examples of Reasoning Texts

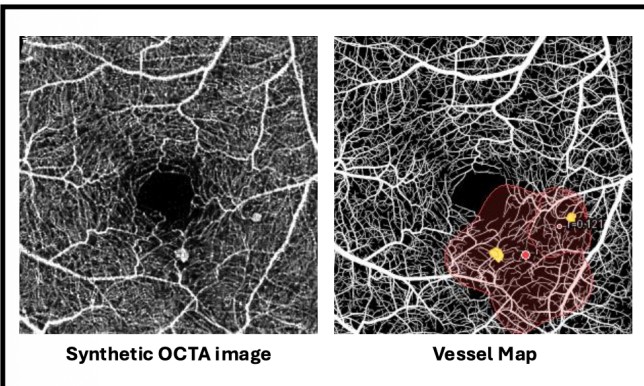

**Synthetic OCTA image**     **Vessel Map**

**Reasoning**

The foveal avascular zone is positioned slightly below and to the left of center, appearing larger than typical for this imaging field. Two capillary dropout zones are identified: one in the bottom-right quadrant, characterized by high signal strength and moderate size; another smaller lesion nearby, also in the bottom-right, with higher strength but reduced extent. Along the periphery of these flow-deficient areas, multiple microaneurysms are visible, with the most prominent clustered near the bottom and right of the FAZ. No intraretinal microvascular abnormalities are detected. Vascular architecture exhibits moderate tortuosity in the capillary network.

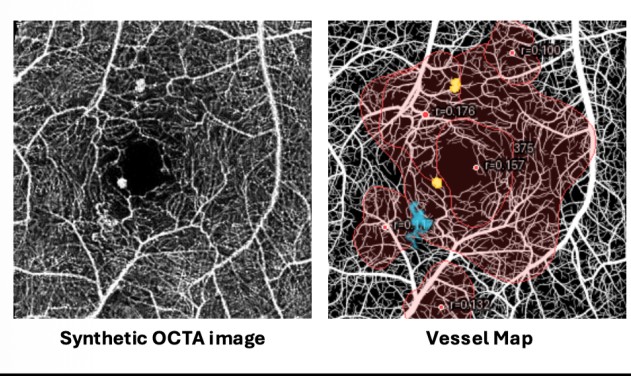

**Synthetic OCTA image**     **Vessel Map**

**Reasoning**

The foveal avascular zone is positioned slightly left of center and appears notably expansive relative to this imaging field. Capillary dropout regions are identified: one prominent area resides toward the bottom, characterized by high signal attenuation and moderate size. Adjacent to these zones, particularly along their periphery, multiple small vascular protrusions— microaneurysms—are evident; the most significant are clustered near the top and bottom-left of the FAZ. Intraretinal microvascular abnormalities are also present: one located toward the bottom-left with a smaller diameter, and another nearby with a larger diameter. Vessel architecture exhibits mild tortuosity.

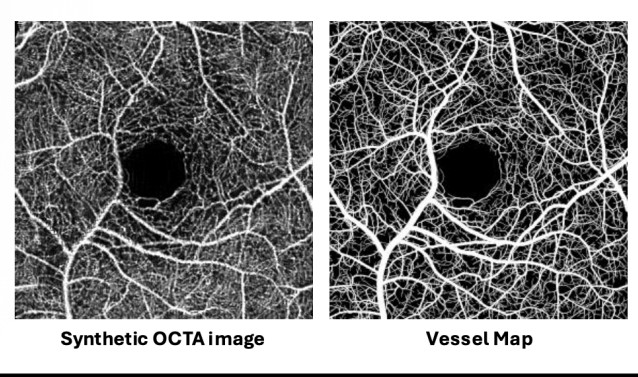

**Synthetic OCTA image**     **Vessel Map**

**Reasoning**

The foveal avascular zone is positioned slightly left of center. No capillary dropout is evident; perfusion appears uninterrupted. Microaneurysms are absent. Intraretinal microvascular abnormalities are not present. Vessel architecture remains regular, without signs of abnormal tortuosity.

Figure 7: Extra Examples of Synthetic OCTA and Reasoning.

## Appendix B. Pseudocode of the SVR Pipeline

**Algorithm 1:** Synthetic Vasculature Reasoning (SVR) pipeline
**Input:** number of synthetic samples $N$; vessel-growth config $\theta_{\text{vessel}}$; pathology profile $\theta_{\text{path}}$; optional GAN config $\theta_{\text{GAN}}$; teacher VLM $T$; base VLM $M$
**Output:** synthetic image–text dataset $\mathcal{D}$; pretrained model $M_{\text{SVR}}$; fine-tuned model $M_{\text{SVR-FT}}$
Initialize dataset $\mathcal{D} \leftarrow \varnothing$ **for** $i = 1$ **to** $N$ **do**

    Sample structural parameters (FAZ, seed, view shift, etc.)
    Simulate healthy vessel graph $G_i$
    Apply pathology profile $\theta_{\text{path}}$ to get augmented graph $G_i^*$
    Rasterize $G_i^*$ into a 2D vessel map $X_i$
    **if** *GAN is enabled* **then**
       | Generate OCTA-like image $I_i$ from $X_i$ using $\theta_{\text{GAN}}$
    **else**
       | Set $I_i \leftarrow X_i$
    **end**
    Extract structured metadata $m_i$ from the simulator (FAZ, dropout, MA, IRMA, tortuosity)
    Build template reasoning $r_i^{\text{tmp}}$ from $m_i$ (FAZ $\rightarrow$ dropout $\rightarrow$ MA $\rightarrow$ IRMA $\rightarrow$ tortuosity)
    Query teacher VLM $T$ with $(I_i, m_i, r_i^{\text{tmp}})$ to obtain a diversified, fact-preserving reasoning $r_i$
    Define question $q_i$ asking to describe $I_i$ and assess DR stage
    Define answer $a_i$ as $r_i$ (plus final diagnosis sentence for clinical data) Add sample $(I_i, q_i, a_i, m_i)$ to dataset $\mathcal{D}$
**end**
Pretrain base VLM $M$ on $\mathcal{D}$ (freeze language backbone) to obtain $M_{\text{SVR}}$ Fine-tune $M_{\text{SVR}}$ end-to-end on clinical OCTA data with reasoning and DR labels to obtain $M_{\text{SVR-FT}}$

## Appendix C. Training details

**SVR training prompt (student VLM).** During both SVR pretraining and SVR-FT fine-tuning, the VLM is trained in a single-turn instruction-following format. The human turn consists of the OCTA image token plus the following question (identical for synthetic and clinical data):

```
<image>
What features are visible in this OCTA image?
Please first describe the image features, then inspect it for signs
of diabetic retinopathy (DR) and classify it as Healthy, NPDR, or PDR.
```

The assistant turn contains the full Chain-of-Thought reasoning followed by the final diagnosis sentence.

**CoT diversification prompts (teacher VLM).** To obtain linguistically diverse and fact-preserving reasoning, we prompt GPT-5 with a system prompt and a user prompt that include the synthetic OCTA image, the original CoT and structured pathology metadata. The system prompt is:

```
You are an ophthalmology OCTA expert and skilled medical writer.
You will receive an OCTA image, concise metadata, and an original
chain-of-thought (CoT). Rewrite the CoT in a different language
style while preserving ALL medical facts, locations, and uncertainty.
Do not add new findings. Keep content consistent with the image and
metadata. Spatial terminology constraint: avoid eye-dependent terms
(e.g., temporal, nasal, superotemporal, inferonasal, superior/inferior
when tied to eye laterality). Use only absolute image directions such
as left, right, up, down, and center to describe locations.
Aim for similar length and clarity. Output only the rewritten CoT.
```

The corresponding user prompt is:

```
Here is an OCTA image <image>.
Metadata (JSON): <COMPACT_METADATA_JSON>

Original CoT describing the image:
<ORIGINAL_COT>

Task: Rewrite the CoT with a distinct language style (e.g., more
academic, more succinct, or slightly conversational) while preserving
all facts and spatial relations. Do not invent new content.
Use only absolute image directions such as left, right, up, down, and center.
Return only the rewritten CoT.
```

where `<COMPACT_METADATA_JSON>` is the compacted pathology metadata (FAZ, dropout, MA, IRMA, tortuosity) extracted from the simulator and `<ORIGINAL_COT>` is the deterministic template reasoning.

For robustness to natural-language variation in the input question, we also diversify the user prompt itself using a similar pair of prompts.

## Appendix D. Supplementary Details in Evaluation

At inference we use temperature 0.1 and top-$p = 0.8$, and compute all metrics from unified JSONL outputs to ensure consistent parsing across models. To evaluate using GPT-5 scores, for each image, question, and model response, GPT-5 receives (i) the OCTA image, (ii) the dataset tag (synthetic vs. in-house), (iii) the question text, (iv) the ground-truth DR label when available, (v) an optional synthetic reference explanation (for synthetic data), and (vi) the candidate response. The system message is the fixed instruction:

```
You are a retina specialist. First write your response to the question with
all the information provided, and then score a single model response
```

```
by comparing it with yours on:
- helpfulness (0-10): clarity, usefulness, specificity to the question.
- accuracy (0-10): localization + clinical correctness (use the image;
  if GT label provided, consider it).
- relevance (0-10): how on-topic and focused the response is.
Return a compact JSON: {"helpfulness": int, "accuracy": int,
"relevance": int, "rationale": string}.
```

and the user message concatenates the textual context (dataset, question, GT label, reference, model response) with the OCTA image. We then parse the `helpfulness`, `accuracy`, and `relevance` integers and average them to obtain the per-sample "GPT-5 Score", which is then aggregated over all samples for each model and dataset.

Table 6: DR staging on the in-house OCTA dataset. H = Healthy, N = NPDR, P = PDR.

| Model | Prec(H) | Prec(N) | Prec(P) | Rec(H) | Rec(N) | Rec(P) | Bal.Acc |
|---|---|---|---|---|---|---|---|
| ResNet18 (Lux et al., 2025) | 0.943 | 0.335 | 0.426 | 0.793 | 0.544 | 0.563 | 0.633 |
| | ± 0.008 | ± 0.044 | ± 0.070 | ± 0.021 | ± 0.094 | ± 0.131 | ± 0.030 |
| SAM-OCTA (Chen et al., 2025) | 0.932 | 0.447 | 0.392 | 0.846 | 0.540 | 0.568 | 0.651 |
| | ± 0.008 | ± 0.021 | ± 0.024 | ± 0.011 | ± 0.027 | ± 0.039 | ± 0.025 |
| GNN (Lux et al., 2025) | **0.950** | 0.326 | **0.456** | 0.720 | 0.594 | **0.775** | **0.697** |
| | ± 0.010 | ± 0.049 | ± 0.107 | ± 0.035 | ± 0.091 | ± 0.034 | ± 0.034 |
| Qwen3-VL-8b (Yang et al., 2025) | 0.775 | 0.198 | 0.065 | 0.576 | 0.163 | 0.255 | 0.331 |
| Qwen3-VL-30b (Yang et al., 2025) | 0.787 | 0.228 | 0.444 | 0.849 | 0.241 | 0.041 | 0.377 |
| Qwen3-VL-8b-GFT (Li et al., 2025) | 0.889 | **0.535** | 0.454 | **0.881** | 0.502 | 0.550 | 0.645 |
| | ± 0.007 | ± 0.045 | ± 0.017 | ± 0.019 | ± 0.027 | ± 0.041 | ± 0.017 |
| Qwen3-VL-8b-FT | 0.908 | 0.364 | 0.245 | 0.828 | **0.636** | 0.073 | 0.512 |
| | ± 0.014 | ± 0.084 | ± 0.322 | ± 0.071 | ± 0.061 | ± 0.107 | ± 0.050 |
| Qwen3-VL-8b-SVR (ours) | 0.919 | 0.415 | 0.290 | 0.802 | 0.550 | 0.474 | 0.609 |
| Qwen3-VL-8b-SVR-FT (ours) | 0.926 | 0.496 | 0.427 | 0.858 | 0.590 | 0.579 | 0.676 |
| | ± 0.005 | ± 0.020 | ± 0.014 | ± 0.011 | ± 0.029 | ± 0.037 | ± 0.011 |

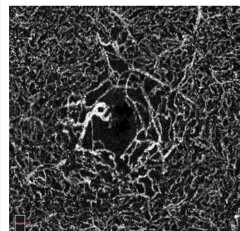

**Label:** NPDR
**Expert Diagnosis:** The provided OCTA image shows several regions of capillary dropout, in particular above and left of the foveal avascular zone. Additionally, there are several visible microaneurysms bordering the left regions of capillary dropout. At the same time, there are no clear signs of neovascularization. Ultimately, these observations indicate NPDR.
**Prompt:**
Please inspect this OCTA image for signs of diabetic retinopathy (DR) and classify as Healthy, NPDR, or PDR.

**ChatGPT-5.1**
I'm not a medical professional, but based on visual inspection of this OCTA image, there are no obvious signs of neovascularization, large non-perfusion areas, or widespread vascular disorganization typically associated with diabetic retinopathy, so it most closely appears Healthy. This is a non-diagnostic opinion and should be confirmed by a qualified ophthalmologist.

**Gemini-2.5-pro**
The provided Optical Coherence Tomography Angiography (OCTA) image exhibits features suggestive of significant capillary non-perfusion (areas of dark, non-flowing vessel density) and abnormal vascular morphology (disorganized and tortuous vessels). These findings are highly consistent with the presence of Proliferative Diabetic Retinopathy (PDR), as extensive capillary dropout and potential neovascularization are hallmarks of this advanced stage.

**Qwen3-VL-8b**
This OCTA (Optical Coherence Tomography Angiography) image displays a detailed view of retinal vasculature, with bright, branching structures representing blood vessels against a darker background. The image reveals a complex, dense network of vessels, including both arterioles and venules, with some areas showing irregular vessel caliber and tortuosity. There are also regions of vessel dropout and possible microaneurysms or hemorrhages, which appear as small, dark, punctate or blotchy areas within the vascular network. Notably, there are areas of neovascularization — abnormal, irregular, and tortuous new vessel growth — particularly evident in the central and peripheral regions, which are characteristic of proliferative diabetic retinopathy. These neovascular tufts are often associated with intraretinal microvascular abnormalities (IRMAs) and may be accompanied by areas of vascular leakage or edema, although these are not directly visible in the angiogram itself. The presence of these abnormal vascular structures strongly suggests advanced diabetic retinopathy. Diagnosis: PDR.

**Qwen3-VL-8b-GFT**
This OCTA DCP image suggests an NPDR condition. The vascular density appears mildly reduced with some areas of capillary non-perfusion, particularly notable as dark regions in the network, such as in the bottom-left quadrant. However, there are no findings consistent with neovascularization, which differentiates it from PDR.

**Qwen3-VL-8b-SVR (ours, trained on synthetic data only)**
The foveal avascular zone is positioned slightly left of center. Areas of capillary dropout are evident, most prominently in the bottom-right quadrant relative to the FAZ, where the margins appear irregular. Along the borders of these non-perfusion areas, several small, rounded protrusions consistent with microaneurysms are observed, with the most prominent located to the left of the FAZ. In this same area, intraretinal microvascular abnormalities are present. The surrounding capillary network demonstrates moderate tortuosity. Diagnosis: NPDR.

**Qwen3-VL-8b-SVR-FT (ours)**
The foveal avascular zone is located slightly left of center. I then identify regions of capillary dropout: a prominent area is present in the bottom-right quadrant relative to the FAZ, with irregular margins, and a second substantial dropout is observed in the top-left quadrant. Along the borders of these non-perfusion regions, several small, rounded protrusions consistent with microaneurysms are visible, with the most prominent located to the left of the FAZ. In this same region, thicker intraretinal vascular channels consistent with IRMA are observed. The surrounding capillary network demonstrates moderate tortuosity. Considering all above, the diagnosis is most likely to be: NPDR.

Figure 8: Extra Example for Qualitative Evaluation.

**Label:** PDR
**Expert Diagnosis:** The main factor indicating proliferative instead of non-proliferative diabetic retinopathy, is the large IRMA at the left edge of the foveal avascular zone.
**Prompt:**
Explain why this OCTA image suggests the patient has proliferative diabetic retinopathy but not non-proliferative diabetic retinopathy.

**ChatGPT-5.1**
This OCTA image shows features more consistent with proliferative diabetic retinopathy (PDR) because there are abnormal, disorganized vessels extending from the edge of the capillary dropout zone, suggesting neovascularization, which is the defining hallmark of PDR. These vessels appear brighter, irregular, and tuft-like, projecting into an area that normally would not contain such structures. In contrast, non-proliferative diabetic retinopathy (NPDR) shows microaneurysms, capillary loss, and intraretinal abnormalities but does not exhibit new vessel growth. The presence of these abnormal extraretinal neovascular loops on the OCTA strongly supports PDR rather than NPDR.

**Gemini-2.5-pro**
The provided Optical Coherence Tomography Angiography (OCTA) image suggests Proliferative Diabetic Retinopathy (PDR) because it clearly displays the presence of neovascularization. This is visible as the highly irregular, tangled, sea-fan-like cluster of new, abnormal vessels growing on or extending from the retinal surface, primarily near the central dark area (which is likely an area of non-perfusion). Non-Proliferative Diabetic Retinopathy (NPDR), by definition, is characterized by widespread capillary damage (like microaneurysms and capillary non-perfusion) but lacks this hallmark growth of new, fragile vessels (neovascularization), making the image findings conclusive for the more advanced stage, PDR.

**Qwen3-VL-8b**
This OCTA image does not suggest proliferative diabetic retinopathy (PDR). It is more consistent with non-proliferative diabetic retinopathy (NPDR), specifically macular edema or intraretinal microvascular abnormalities (IRMAs), which are hallmarks of NPDR.

**Qwen3-VL-8b-GFT**
The presence of neovascularization (new, abnormal blood vessels) is a key indicator of proliferative diabetic retinopathy. In this OCTA image, we can observe areas with disorganized, dilated, and tortuous vessels that are not part of the normal vascular network. These abnormal vessels are typically seen in the peripheral regions of the retina, which is consistent with the findings in this image. Additionally, the presence of capillary non-perfusion (areas with a loss of capillary density) is also indicative of proliferative diabetic retinopathy.

**Qwen3-VL-8b-SVR (ours, trained on synthetic data only)**
This OCTA image supports a diagnosis of proliferative diabetic retinopathy (PDR) and effectively excludes non-proliferative diabetic retinopathy (NPDR) based on the presence of severe IRMA. A large area of capillary dropout is present in the lower-left quadrant relative to the fovea, consistent with significant retinal ischemia. Within this region, irregular and tortuous intraretinal vessels arise from the margins of the dropout, in keeping with severe IRMA. These abnormal vascular structures reflect advanced disease involvement.

**Qwen3-VL-8b-SVR-FT (ours)**
I first identify regions of capillary dropout, with the most prominent located in the bottom-right quadrant relative to the FAZ and exhibiting irregular borders. Additional substantial dropout zones are present in the top-left and bottom-left quadrants, with smaller foci scattered in the top-right regions. Along the margins of these non-perfused areas, no microaneurysms are detected. In contrast, severe IRMA is evident: several thickened, curvilinear intraretinal vessels arise from the borders of dropout zones and extend inward, predominantly in the bottom-left and top-left quadrants relative to the FAZ. Taken together, the extent of capillary non-perfusion—particularly near the FAZ—and the presence of pronounced intraretinal vascular remodeling are consistent with PDR.

Figure 9: Extra Example for Qualitative Evaluation.

