# OpenReview forum: "Synthetic Vasculature and Pathology Enhance Vision-Language Model Reasoning"
_MIDL.io/2026/Conference — MIDL 2026 Poster_

### Official Review · Reviewer_GQCk · 2026-01-10

**Confidence:** 2
**Preliminary Rating:** 4
**Final Rating:** 5

**Summary:**

The papers contributions are in 3 folds: (1) It extends available OCTA vessel graph generation algorithms specific to healthy eyes to DR related pathologies. (2) A synthetic dataset consisting of generated OCTA images and paired text with reasoning and disease related localization. (3) An open source VLM is finetuned further with the curated synthetic dataset to improve the text generation for clinical OCTA images.

**Strengths:**

* The synthetic dataset and code are publicly available. But currently the dataset has restricted access.
* They extended the available algorithmic vessel graph generation specific to healthy eyes to DR pathologies
* Using CoT, the proposed VLM has better explainability and disease related localization

**Weaknesses:**

* Single disease is very limiting due to disease specific algorithmic pathology simulation. It is not possible to easily extend the proposed solution to another retinal disease.
* The proposed GPT-5 score lacks formalities and guarantees that it is an objective assessment metric. Especially the localization aspect is only evaluated with GPT-5. From this perspective, the paper reads as a distillation approach from GPT-5 to another VLM
* In Table 1 for DR classification, instead of Resnet18, there should be comparisons against better architectures ideally with OCTA pretraining [1]
* It is not clear whether SVR-FT has its text model finetuned as well.


References

[1] Chen, Xinrun, et al. "Sam-octa: Prompting segment-anything for octa image segmentation." Biomedical Signal Processing and Control 106 (2025)

**Detailed Comments:**

Please see Weaknesses and Questions To Address In The Rebutta

**Justification Of Final Rating:**

Authors clarified my questions, and included an evaluation study by clinicians showing that gpt based evaluation correlates highly with the clinicians. It strengthened the protocol used for evaluating synthetic dataset

**Justification Of The Preliminary Rating:**

The paper proposes to generate a large scale OCTA image and paired text with reasoning and localization in order to finetune available VLMs. They showed strong improvement in DR classification with specific pathologies explained in the generated text report. There are issues with evaluation, such as over-relying on GPT5 to give scores.

**Questions To Address In The Rebuttal:**

* Could authors explain whether GAN has topological guarantees or not?
* If GPT-5 is good to use as evaluator, what is the need for generating a synthetic dataset and finetuning another VLM?
* If SVR-FT’s text model is finetuned (it is not clear in the manuscript), high GPT-5 score can be explained by learning to match its style rather than better quality. Especially localization aspect should be evaluated with more objective and meaningful metrics, ideally by clinicians
* Also, please address the issues mentioned in the weaknesses

---

> ### Author Response · Authors · 2026-01-23
> **Response to Reviewer GQCk**
>
> We thank the reviewer for the recognition of the technical motivation and the value of the synthetic dataset. Below we respond to the concerns.
>
> **(1) Dataset access.** We apologize, this was due to an issue with the data-sharing platform and has been resolved.
>
> **(2) Extensibility.** We agree focusing on one disease is a limitation. While targeting DR, the pipeline is designed to be parameter-driven. The simulation controls remodeling via explicit parameters (e.g., oxygen sinks), tunable for other pathologies. We added a discussion on extending the framework to diseases like RVO and Glaucoma **(Sec. 4.3)**.
>
> **(3) Use of teacher VLM and GPT-5 scores.** We fully agree that evaluating generative reasoning remains a known challenge in the field, and that GPT-5–based scores alone are insufficient to guarantee clinical correctness. In the revision, we explicitly acknowledge this limitation and clarify that GPT-5 scores are not treated as a ground-truth clinical metric, but rather as an auxiliary metric, which is inspired by a similar setup in the visual instruction tuning (LLaVA) paper [1], which is widely cited and foundational in the multimodal instruction following literature. In order to study how correlated the GPT-5 scores are to clinical truth, we carried out a correlation analysis between GPT-5 score, classification performance, and expert ratings **(Sec. 4.2 and Fig. 4)**. We observe a strong alignment in the experiments, indicating that while imperfect, GPT-5 scores can provide a meaningful proxy when interpreted alongside expert judgment rather than in isolation. This proxy is especially useful for us when training and evaluating models in the developmental phase because we cannot run expert rating experiments for all development stages.
>
> Additionally, we clarify that our approach is not distillation from GPT-5: In the data generation process, GPT-5 is not used to produce pathology labels or “ground-truth” diagnoses. All pathology presence, location, and attributes are determined by the simulator metadata. GPT-5 is used only for language-style rewriting **(Sec. 2.3.1)**.
>
> **(4) Stronger baseline (SAM-OCTA).**  Following the suggestion, we added a SAM-OCTA–based baseline by fine-tuning its pretrained weights with an MLP head **(Tabs. 1 & 6 / Sec. 3.2)**.
>
> **(5) Clarifying whether SVR-FT fine-tunes the text model.** Thank you for pointing out the ambiguity. We have revised the manuscript to explicitly specify that the language module is not fine-tuned, only the vision-side and cross-modal components are updated. This removes ambiguity and addresses the concern that improved GPT-5 scores could be explained by learning to mimic linguistic style. Together with the newly introduced objective localization metric (# of correct locs.) and the expanded expert evaluation, we believe that our evaluation now better reflects clinical accuracy and relevance but not linguistic fluency **(Secs. 3.3 & 4.2 / Tab. 4)**.
>
> **(6) GAN topological guarantees.** Our generator does not provide formal mathematical topological guarantees in the strict sense. Instead, vessel topology is explicitly defined by the biologically inspired growth and perfusion simulation, which enforces physiologically plausible connectivity, branching patterns, and remodeling under controlled parameters. The GAN component is used only as a post-processing module to add realistic appearance variations (e.g., noise and imaging artifacts) without modifying the underlying vessel graph. In practice, these textural changes do not approach the magnitude of structural alteration caused by pathology. During training, the GAN is supervised with segmentation-based losses to preserve structural fidelity, following prior OCTA synthesis work [2].
>
> **(7) "If GPT-5 is used, why generate synthetic data and fine-tune another VLM?"** We appreciate this question and clarified the motivation. GPT-5 is not used as a clinical diagnostic tool and is not a deployable target in many practical settings (cost, privacy, reproducibility, integration, standalone performance). The purpose of SVR is to create structured, scalable synthetic supervision so that open-source VLMs can be trained to produce clinically grounded explanations and improve DR staging (validated by classification performance, expert ratings, and localization metrics). GPT-5 serves only as a controlled rewriting tool and an auxiliary evaluator, whereas the main gains come from the simulator-grounded supervision and dataset scale **(Secs. 2.3.1 & 4)**.  Finally, we emphasize that general-purpose VLMs alone fail to resolve these fine-grained pathological features, as shown by the near-chance performance of the non-finetuned baselines in our study.
>
> **References**
>
> [1] Liu, Haotian, et al. "Visual instruction tuning." NeurIPS (2023).
>
> [2] Kreitner, Linus, et al. "Synthetic optical coherence tomography angiographs for detailed retinal vessel segmentation without human annotations." IEEE TMI (2024).

---

> ### Comment · Reviewer_GQCk · 2026-01-26
>
> Thanks for the clarifications.The expert evaluation is crucial in my opinion and i dont have any further question. I will raise my score after the discussion period.

---

> > ### Author Response · Authors · 2026-01-26
> >
> > Thank you for your feedback. We are pleased to hear that our rebuttal and the expanded evaluation resolved your questions. We appreciate your support and your decision to reconsider the rating.

---

### Official Review · Reviewer_gdPj · 2026-01-12

**Confidence:** 4
**Preliminary Rating:** 3
**Final Rating:** 4

**Summary:**

This paper introduces Synthetic Vasculature Reasoning (SVR), a framework for training vision–language models using fully synthetic Optical Coherence Tomography Angiography (OCTA) images paired with synthetic, pathology-grounded chain-of-thought reasoning. The results show that fine-tuning a general-purpose VLM on this synthetic data enables strong zero-shot diabetic retinopathy classification.

**Strengths:**

* This research has strong motivation from both technical and clinical perspectives.
* The experiments and results are well structured and easy to follow.
* The 100k synthetic dataset could significantly benefit research in this area.

**Weaknesses:**

A key concern is the evaluation metrics. While the paper reports improvements in both classification performance and explanation quality, the latter relies heavily on GPT-5–based scores, which do not necessarily reflect true clinical correctness and real-world applicability. Although the Expert Rating is provided, but the protocol of the rating protocol is not provided in the paper.

**Detailed Comments:**

* A clear analysis of failure cases where GPT-5 scores and expert judgments disagree would strengthen confidence in the evaluation.

**Justification Of Final Rating:**

I would like to thank the authors for thoroughly addressing my concern regarding model validation by providing additional statistical analysis and providing a case when the model failed. I am happy to increase the final rating accordingly.

**Justification Of The Preliminary Rating:**

This research presents a novel framework for training VLMs using synthetic data. However, the model evaluation could be strengthened to better demonstrate the intended purpose and clinical value of the proposed VLM.

**Questions To Address In The Rebuttal:**

Please address the concerns in weakness and detailed comments.

---

> ### Author Response · Authors · 2026-01-23
> **Response to Reviewer gdPj**
>
> We thank the reviewer for raising important concerns regarding the evaluation methodology and clinical validity.
>
> **(1) Use of GPT-5 scores.** We fully agree that GPT-5–based scores alone are insufficient to establish clinical correctness. In the revision, we explicitly acknowledge this limitation and clarify that GPT-5 scores are not treated as a ground-truth clinical metric, but rather as an auxiliary metric, which is inspired by a similar setup in the visual instruction tuning (LLaVA) paper [1], which is widely cited and foundational in the multimodal instruction following literature. In order to study how correlated the GPT-5 scores are to clinical truth, we carried out a correlation analysis between GPT-5 score and classification performance, we now move this correlation analysis to the main manuscript. Additionally, we also add a correlation analysis between GPT-5 scores and expert ratings **(see Sec. 4.2 and Fig. 4)**. We observe a strong alignment in both experiments, indicating that while imperfect, GPT-5 scores can provide a meaningful proxy when interpreted alongside expert judgment rather than in isolation. This proxy is especially useful for us when training and evaluating models in the developmental phase because we cannot run expert rating experiments for all development stages.
>
> **(2) Expert rating protocol and objective localization.** We expanded the description of the expert rating protocol, the evaluation criteria, rating scale, and review procedure used by the experts. To further reduce reliance on LLM judgment, we expanded the expert rating experiment and introduced an objective localization metric (number of correct localizations per case), which measures whether explanations correctly reference clinical regions. This provides a concrete comparison of the models’ clinical grounding **(see Sec. 3.2 and Tab. 4)**.
>
> Following the reviewer’s suggestion, we added an explicit failure case analysis, presenting representative examples where GPT-5 scores and expert judgments disagree. These cases highlight that GPT-5 may overestimate explanations with fluent language but imperfect localization, reinforcing the need for expert review and objective metrics. Beyond higher GPT-5 scores, SVR-based models also achieved better expert ratings and localization accuracy, indicating that the performance gains reflect actual clinical utility rather than the model simply learning to copy a writing style **(see Sec. 4.2 and Fig. 6)**.
>
> Lastly, we would like to emphasize that the goal of this work is to explore the methodological feasibility of using biologically-inspired simulation and large-scale synthetic data to effectively finetune VLMs for explainable diagnosis, and we have shown that our models outperformed all the other VLM baselines in terms of classification performance. Moving forward, we plan to validate these findings in larger-scale clinical studies with broader expert participation and disease coverage.
>
> **Reference**
>
> [1] Liu, Haotian, et al. "Visual instruction tuning." Advances in neural information processing systems 36 (2023): 34892-34916.

---

### Official Review · Reviewer_7XxM · 2026-01-16

**Confidence:** 4
**Preliminary Rating:** 5
**Final Rating:** 5

**Summary:**

The paper proposes Synthetic Vasculature Reasoning (SVR), a framework that generates pathology-aware synthetic OCTA images along with grounded, diversified chain-of-thought explanations derived from simulator metadata. They use this synthetic image–reasoning dataset, OCTA-100K-SVR, to pretrain a VLM and then fine-tune on limited real OCTA data. They achieve stronger DR staging performance and more clinically meaningful explanations with scaling results, ablations, and domain expert and LLM-based evaluation.

**Strengths:**

- The paper is very well-written and well-structured, presenting clear evaluation results to support their proposed method.
- The evaluation is thorough. They run experiments on two real clinical datasets (public and in-house datasets), look at zero-shot and fine-tuning. The scaling and ablation study makes a convincing case that the key components actually matter.
- Beyond just reporting accuracy, they explicitly evaluate explanation quality, including localization and clinical correctness, using both LLM judging and domain expert review, showing improvements in clinically meaningful reasoning. I especially like this part, which makes the work clinically practical.

**Weaknesses:**

- Figure 1 is generally a good figure showing the whole framework, but it’s a bit hard to read due to visual density. I would recommend separating the three stages: training, reasoning, and zero-shot, with distinct background shading or boxed regions (and clearer headings).
- The paper uses GPT-5 as the teacher VLM, but doesn’t really say why. A short explanation of the choice would be helpful.

**Detailed Comments:**

- Same questions already mentioned above: why was GPT-5 selected as the teacher VLM instead of other models? How sensitive is the result to the teacher model?

**Justification Of Final Rating:**

Thank the authors for the rebuttal. The authors have answered all of my questions and resolved the concerns raised in my review. I am maintaining my score. The manuscript remains highly convincing and well-justified.

**Justification Of The Preliminary Rating:**

The paper is well-written and well-structured, presenting clear experiments that support the proposed method’s effectiveness. I only have some minor concerns regarding the figures and the choice of the teacher model. I especially adore the completeness of the evaluation, as the author brings in the ophthalmology expert review into consideration, making this work clinically meaningful.

**Questions To Address In The Rebuttal:**

Please see the “weakness” and “detailed comments” sections.

---

> ### Author Response · Authors · 2026-01-23
> **Response to Reviewer 7XxM**
>
> We thank the reviewer for the positive assessment of our work and for the constructive suggestions. We are glad that the reviewer found the paper well written, the method effective, the evaluation thorough, and the explanation-quality analysis clinically meaningful. Below we address the specific comments.
>
> **(1) Fig. 1 readability.**
> In response to the reviewer’s suggestion, we revised Fig. 1 to improve readability by visually separating the pipeline into clearly distinguishable components. Specifically, we added background shading to highlight the multi-step reasoning module and the zero-shot evaluation stage, and made minor adjustments to layout, spacing, and element positioning to reduce visual density.
>
> **(2) Choice and role of GPT-5 as the teacher VLM.**
> We thank the reviewer for raising this point and have clarified the role and motivation of GPT-5 in the revised manuscript **(see Sec. 2.3.1)**. GPT-5 is used **only for language-style rewriting**, rather than as a diagnostic tool or source of pathology labels. All pathological structures, locations, and disease states are determined by the simulator and its metadata, GPT-5 does not introduce any new clinical content.
>
> We chose GPT-5 primarily due to its **strong instruction-following capability, stable long-form generation, and reliable adherence to structured prompts**, which are important for rewriting simulator-grounded explanations into fluent, diverse, and clinically coherent reasoning text without altering the underlying pathology. In practice, we found GPT-5 to be more consistent and less prone to hallucinating or deviating from provided constraints compared to alternative models we tested in our experiments, making it a reliable choice for controlled language rewriting.

---

### Author Rebuttal · Authors · 2026-01-23

**Rebuttal:**

We thank all reviewers for their careful reading and constructive feedback. In the revision we propose the following changes:

**(C1) Fig. 1 readability:** We improved Fig. 1 by adding background shading to visually distinguish the multi-step reasoning and zero-shot evaluation components, as suggested, and by making minor layout and positioning adjustments to reduce visual density. **[Fig. 1]**

**(C2) Stronger baselines:** We expanded the DR classification baselines to include a stronger OCTA-pretrained architecture (SAM-OCTA–based), addressing concerns about reliance on ResNet18 and the GNN approach alone. This provides a more competitive and clinically relevant comparison. **[Sec. 3.2 / Tabs. 1 & 6]**

**(C3) Improved expert evaluation and localization metrics:** We expanded the description of the expert rating protocol to clarify evaluation criteria and procedures, and introduced an objective localization metric (# of correct locs.). **[Secs. 3.2 & 4.2 / Tab. 4]**

**(C4) GPT-5 score rationale:** We further clarified the motivation for adopting GPT-5 as an auxiliary evaluation tool and studied the correlation between GPT-5 scores and model performance in terms of classification performance and expert ratings. We also added an explicit disagreement analysis between GPT-5 scores and expert judgments. **[Sec. 4.2 / Figs. 4 & 6]**

**(C5) Clarification of fine-tuning process:** We explicitly specify which components are fine-tuned in SVR-FT, eliminating ambiguity in the training process. **[Sec. 3.3]**

**(C6) Discussion on framework extensibility:** We added a new section discussing the generalization of the SVR framework. We explain how the simulator can be retuned for other diseases and outline future directions. **[Sec. 4.3]**

**(C7) Clinical terminology refinement:** We refined the terminology in the pathology generation. Instead of using the general term neovascularization (NV) to describe abnormal vascular changes, we use the more specific term intraretinal microvascular abnormalities (IRMA) throughout the synthetic generation, model training, and inference stages. This change reflects only a refinement of clinical terminology rather than a methodological modification. Accordingly, the results for the SVR and SVR-FT models remain consistent, with only minor numerical or textual adjustments.

All changes are highlighted in dark blue. We hope for a fruitful discussion and are more than happy to elaborate on any further questions.

**Supporting Material:**

/attachment/e4e9ade618b02752014f7292d9c0ea529ca4f45e.pdf

---

### Meta-Review · Area_Chair_FmCy · 2026-02-04

**Recommendation:** Accept (Poster)
**Confidence:** 5

**Metareview:**

This paper introduces Synthetic Vasculature Reasoning (SVR), a framework that leverages synthetic OCTA images and grounded explanations to improve Vision-Language Model performance in diabetic retinopathy diagnosis. Reviewers highlighted the value of the generated OCTA-100K-SVR dataset and the model's strong zero-shot and fine-tuned results. Although concerns were initially raised regarding the use of GPT-5 as an evaluator and the strength of baselines, the authors effectively addressed these by demonstrating a high correlation between GPT-5 scores and expert ratings, adding objective localization metrics, and including a SAM-OCTA baseline. Additionally, the revised manuscript clarifies the training procedure and discusses the framework's potential for other retinal diseases. Given the significant methodological contribution and rigorous validation, I recommend accepting this paper for MIDL 2026.

---

### Decision · Program_Chairs · 2026-02-13

Accept (Poster)